# Fast Fourier Convolutions in Self-Supervised Neural Networks for Image Denoising

**Joonas Ariva & Mikhail Papkov**
Institute of Computer Science
University of Tartu
`{joonas.ariva, mikhail.papkov}@ut.ee`

## Abstract

Recently, denoising convolutional neural networks (CNN) have started to outperform classical denoising algorithms. However, CNNs performance could be constrained by the limited receptive field of regular convolution. To mitigate this problem, a new modification for CNNs was proposed: Fast Fourier Convolution (FFC). Here, a global receptive field is achieved by using Fourier Transform and convolving spectral representation. The global perception field can help CNNs to better capture dependencies in image regions that are far apart. In this work, we design multiple approaches for incorporating FFC into self-supervised neural networks for image denoising. We evaluate these approaches on three benchmark datasets and compare them with supervised and self-supervised methods. We empirically show that an FFC-enhanced denoising network achieves the state-of-the- art results on the character dataset and shows a comparable level of performance for both grayscale and color natural images.

## 1 Introduction

Chi et al. (2020) proposed Fast Fourier Convolution (FFC) with non-local receptive field. Unlike a regular convolution operating locally, FFCs can capture essential details of the image far apart in the spectral domain. FFC blocks can be seamlessly imputed in place of vanilla convolutions.

An FFC block is divided into two branches with local and global receptive fields. The input tensor is split channel-wise between the two paths (usually in halfs). The local branch works as a normal convolutional layer. The global branch operates in the spectral domain and carries out global updates to the input.

Multiple state-of-the-art neural networks could benefit from the use of FFC for image recognition, super-resolution (Zhang et al., 2022), segmentation (Farshad et al., 2022), large image inpainting (Suvorov et al., 2022), etc. In this work, we integrate FFCs in a self-supervised denoising framework, Noise2Same (Xie et al., 2020). We conduct an ablation study for the optimal FFC position in the U-Net (Ronneberger et al., 2015) backbone and empirically find suitable data domains for FFC application.

## 2 Architecture

We follow Xie et al. (2020) for the baseline U-Net implementation and explore different strategies to replace normal convolutions with FFC. We aim to keep our implementations similar to the baseline in terms of the number of parameters and overall U-shape for comparability.

FFC blocks are only reasonable to use if at least three of them are in a row (the first and last blocks are used for splitting/merging local and global channels, and middle blocks perform spectral convolutions). This conflicts with the original U-Net design with two convolutional blocks per level. We explore three different FFC U-Net designs that solve this problem.

In **Vanilla** design, we switch U-Net's down- and upsampling convolutions also with FFC blocks to provide us with three FFC blocks for each level. In **Extra Layer** design we keep the regular down- and upsampling convolutions but add an extra convolutional layer to each level. In **Twin Pass** design, local and global channels are not merged together at the end of every level. Both spatial and spectral information flow through the network in parallel and so we do not need to worry about the number of convolutional blocks in a level. All designs are summarized in Figure 4 and can be used in U-Net's encoder/decoder (or both).

## 3 EXPERIMENTS

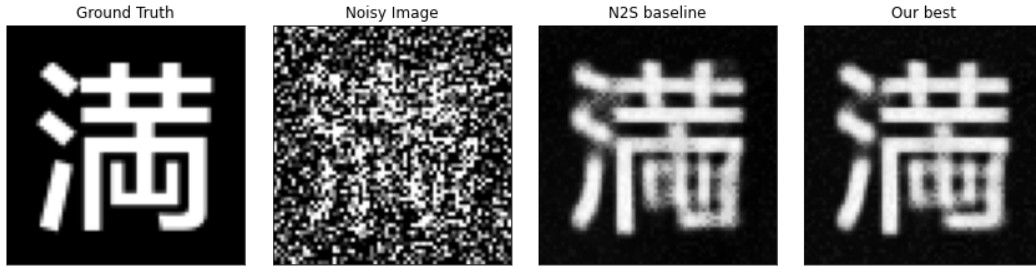

Figure 1: Visualization of testing results on HànZì dataset. We compare our best model (extra layer design with FFC encoder and decoder) against vanilla Noise2Same baseline. We see that in our model, the character is "diffusing" less to the background.

Table 1: Comparisons among denoising methods on different datasets, in terms of Peak Signal-to-Noise Ratio (PSNR). Best metrics are highlighted in bold. The extended table is in appendix.

|  | Methods | Datasets | | |
|  |  | ImageNet | HànZì | BSD68 |
|---|---|---|---|---|
|  | Input | 9.69 | 6.45 | 20.19 |
| *Self-Supervised* | Noise2Self-Donut [3] | 8.62 | 13.29 | **28.20** |
|  | Noise2Same [2] | 22.26 | 14.38 | 27.95 |
|  | Noise2Same (our implementation) | **22.84** | 14.83 | 28.14 |
|  | Extra Layer (FFC encoder & decoder) | 22.65 | **15.72** | 28.13 |
|  | Twin Pass (FFC encoder) | 22.34 | 15.28 | 28.16 |

We tested these designs on two natural image datasets, BSD68 (Martin et al., 2001) and ImageNet (Deng et al., 2009), and also on the Chinese character dataset HànZì (C.-L. Liu, 2011). The image denoising setup used is based on the setup used in Noise2Same paper (Xie et al., 2020).

We compare our best models with other methods in Table 1 and summarize our own results in Table 2. Denoising examples for each dataset are shown in Figures 1, 2 and 3. On BSD68, FFC models match baseline results or slightly exceed them. On ImageNet our model's performance slightly declines compared to the baseline. We compare our best-performing models to other denoising models in Table 1. We see that on HànZì our models beat current state-of-the-art models including supervised models (Table 3).

## 4 CONCLUSION

We tested three different FFC neural network designs for image denoising on three datasets. We observed state-of-the-art results on HànZì and comparable level of performance on natural image datasets. Among our model designs, no design was superior in all experiments. Our results suggest that FFC can improve denoising performance in some cases, and this topic is worth exploring further.

URM STATEMENT

The authors acknowledge that at least one key author of this work meets the URM criteria of the ICLR 2023 Tiny Papers Track.

ACKNOWLEDGEMENTS

We thank Revvity Inc. for support and High Performance Computing Center of the Institute of Computer Science at the University of Tartu for the provided computing power.

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

## A  TRAINING SETUP

All neural networks were implemented in PyTorch Paszke et al. (2019) and were trained on Tesla V100-PCIE-16GB or V100-SXM2-32GB GPUs. Training consisted of 80 000 iterations for BSD68 and 50 000 iterations for other datasets. We used batch size of 64 for each dataset and Adam optimizer (Kingma & Ba, 2014) with learning rate of $4 \times 10^{-4}$. Our code is available at `https://github.com/JoonasAriva/noise2same.pytorch`.

For models where U-Net's skip connection would merge global channels to local or vice versa, we added a $1 \times 1$ convolution so that model could better learn how to mix these channels. To address deviations from the *Noise2Same* design, we also added different baseline models with an extra layer in residual block or with $1 \times 1$ skip convolution (Table 2). The ratio of global channels $\alpha$ was fixed to 0.5 for the FFC models.

## B  DENOISING EXAMPLES

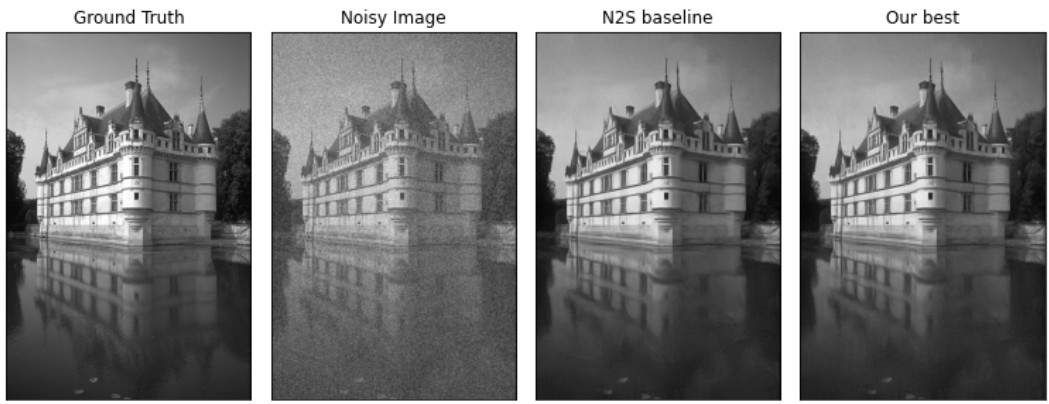

Figure 2: Visualization of testing results on BSD68 dataset. We compare one of our best models (twin pass without FFC decoder) against vanilla *Noise2Same* baseline. Visually, results look identical.

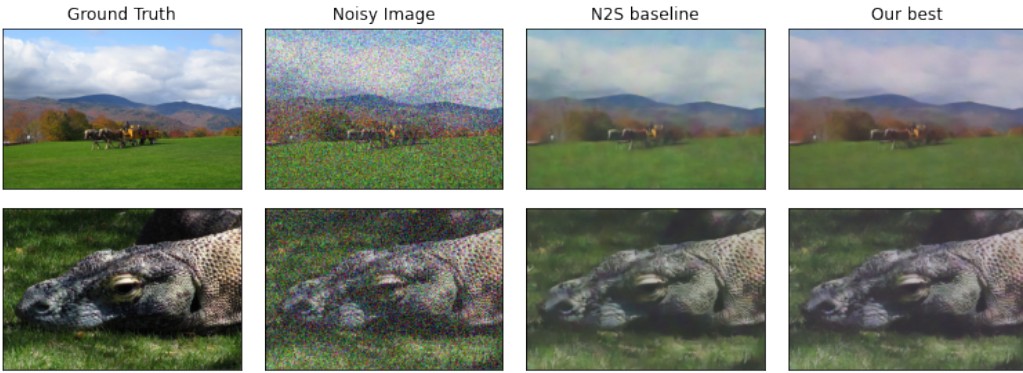

Figure 3: Visualization of testing results on Imagenet dataset. We compare our best model (extra layer design with FFC encoder and decoder) against vanilla *Noise2Same* baseline. In some cases, FFC models tend to emphasise the red channel of the image while denoising. We hypothesise that models with FFCs might have problems adapting to multichannel images.

## C  ABLATION STUDY & EXTENDED COMPARISON WITH OTHER METHODS

Table 2: Ablation study among different denoising FFC designs on different datasets, in terms of Peak Signal-to-noise Ratio (PSNR). Bold numbers indicate best performance for the dataset. Checkmarks indicate what components are present in the network. Different designs are ordered into three groups: designs which don't need $1 \times 1$ skip convolution, designs which need it and finally extra layer designs (also no need for $1 \times 1$ convolution). Baseline for each group is in top row. Parameter count and average inference time for a single $64 \times 64$ image is also given for networks.

| ID | FFC encoder | FFC decoder | extra layer | twin pass | 1x1 skip conv | ImageNet | Hanzi | BSD68 | Params | Inference time (ms) |
|----|----|----|----|----|----|----|----|----|----|----|
| 1 | | | | | | 22.84 | 14.83 | 28.14 | 5.8M | 3.68 |
| 2 | ✓ | | | | | 22.34 | 15.19 | **28.16** | 5.5M | 10.04 |
| 3 | ✓ | ✓ | | ✓ | | 22.19 | 15.21 | 28.13 | 4.4M | 17.38 |
| 4 | | | | | ✓ | 22.81 | 14.84 | 28.13 | 6.0M | 5.35 |
| 5 | ✓ | ✓ | | | ✓ | 22.49 | 15.18 | **28.16** | 5.4M | 14.20 |
| 6 | ✓ | | | ✓ | ✓ | 22.34 | 15.28 | **28.16** | 5.3M | 11.46 |
| 7 | | | ✓ | | | **23.06** | 15.42 | **28.16** | 7.9M | 6.39 |
| 8 | ✓ | ✓ | ✓ | | | 22.65 | **15.72** | 28.13 | 7.6M | 13.67 |
| 9 | ✓ | | ✓ | | | 22.39 | 15.04 | 28.09 | 7.6M | 9.71 |
| Noisy input | | | | | | 9.69 | 6.45 | 20.19 | - | - |

Table 3: Comparisons among denoising methods on different datasets, in terms of Peak Signal-to-Noise Ratio (PSNR). *Noise2Self-Noise* and *Noise2Self-Donut* refer to two masking strategies mentioned in Batson & Royer (2019), where the original results presented in Batson & Royer (2019) are produced by the noise masking. Bold numbers indicate the best performance among self-supervised methods.

| | | Datasets | | |
|----|----|----|----|----|
| | **Methods** | ImageNet | HànZì | BSD68 |
| *Traditional* | Input | 9.69 | 6.45 | 20.19 |
| | NLM Buades et al. (2011) | 18.04 | 8.41 | 22.73 |
| | BM3D Dabov et al. (2007) | 18.74 | 10.90 | 28.59 |
| *Supervised* | Noise2True | 23.39 | 15.66 | 29.06 |
| | Noise2Noise Lehtinen et al. (2018) | 23.27 | 14.30 | 28.86 |
| *Self-Supervised* | Noise2Void Krull et al. (2019) | 21.36 | 13.72 | 27.71 |
| | Noise2Self-Noise Batson & Royer (2019) | 20.38 | 13.94 | 26.98 |
| | Noise2Self-Donut Batson & Royer (2019) | 8.62 | 13.29 | **28.20** |
| | Noise2Same Xie et al. (2020) | 22.26 | 14.38 | 27.95 |
| | Noise2Same (ours) | **22.84** | 14.83 | 28.14 |
| | Extra Layer (enc&dec) | 22.65 | **15.72** | 28.13 |
| | Twin Pass (enc) | 22.34 | 15.28 | 28.16 |

## D FFC DESIGNS VISUALIZED

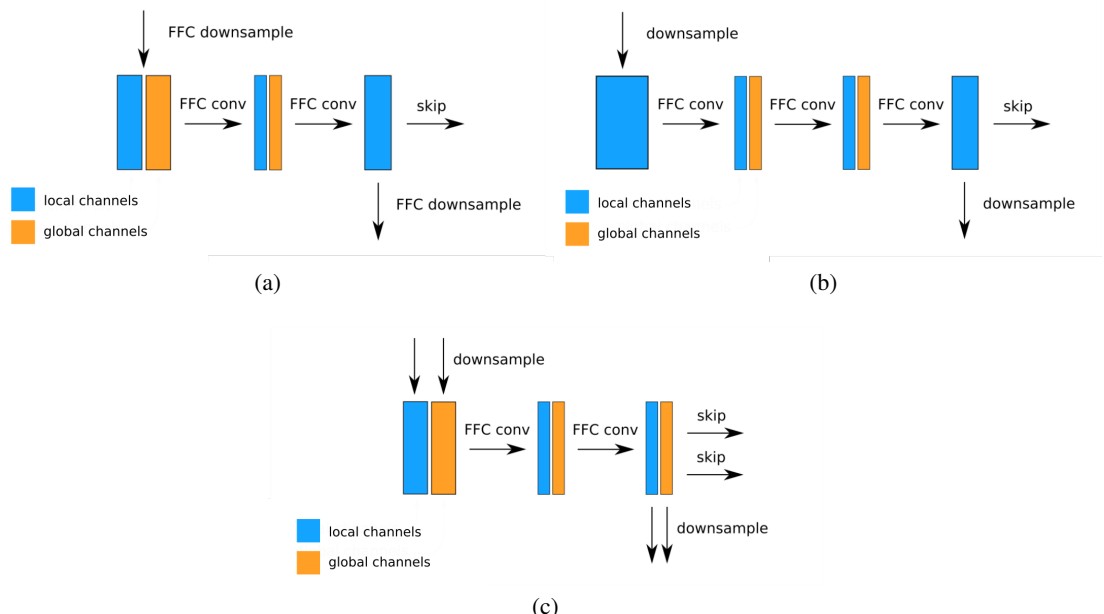

Figure 4: (a) Vanilla FFC design. (b) Extra Layer design. (c) Twin Pass design. All are modified residual blocks from U-Net encoder. Decoder designs are mirrored. Residual block shortcuts are not included in the figures.

