# OpenReview forum: "Fast Fourier Convolutions in Self-Supervised Neural Networks for Image Denoising"
_ICLR.cc/2023/TinyPapers — Submitted to Tiny Papers @ ICLR 2023_

### Official Review · Reviewer_8Jkr · 2023-03-31

**Confidence:** 4

**Summary Of Contributions:**

The authors proposed to use Fast Fourier convolution for image denoising task as for image denoising local receptive field may not be able to get the global dependencies of the masked and noisy pixels. The authors study different approaches to incorporate FFC into image denoising framework. TO the end, all three variants are tested against standard image denoising approaches on three image datasets.

**Rating:**

Clear, Correct, and Reproducible (CCR): a submission which meets the reviewing criteria

**Strengths And Weaknesses:**

- Clarity: The paper was easy to follow through and relevant related works are cited.

- Correctness: The main claims "We empirically show that an FFC-enhanced denoising network achieves the state-of-the-art results on the character dataset and shows a comparable level of performance for both grayscale and color natural images." are justified by the findings. However, the performance gains are marginal and maybe the side effect of different initial seeds.

- Reproducibility: No code is provided but I see enough details in Appendix to reproduce the analysis and datasets seems to be standard in the computer vision community.

- Follows basic requirements: yes

**Suggested Changes:**

- Experiments:
    -  Re-run experiments a few times and report average scores with variance.
    -  Given that the performance gain is marginal it is important to conduct a statistical significance test.

---

### Official Review · Reviewer_HGU4 · 2023-04-01

**Confidence:** 3

**Summary Of Contributions:**

The paper proposes the use of Fast Fourier Convolutions (FFC) to obtain a global receptive field and improve standard denoising CNNs. The proposed method is compared with U-Net-based baselines on three public datasets of various types.

**Rating:**

High Potential (HP): a submission which meets the reviewing criteria and has potential to make an impact on the field

**Strengths And Weaknesses:**

### Strengths
- The proposed approach is of interest and can have a significant impact on the denoising community.
- The authors' effort in making the comparison with other methods as fair as possible is worthy.
- The authors describe and evaluate three different variations of the proposed approach.
- The supplementary material includes valuable additional information, such as the model architecture, an ablation study, results from additional competitors.
- Detailed information about the architecture and training setup are provided, making the approach reproducible.
- Overall, the paper is technically sound and convincing. It is well written and easy to follow, and the design choices are justified and explained clearly.

### Weaknesses
In my opinion, the paper has one main weakness: the motivation behind the use of FFCs instead of CNNs. It is stated that FFCs solve the CNN problem of having local, limited receptive field. However, this is true for a local convolution operation while convolutional layers at later stages of a deep model usually have quite wide receptive field. This interpretation seems justified by the results of Table 1 and 2, where it is shown that the performance of CNN and FFC-based methods are similar.

Other minor weaknesses are:
- The only metric used to compare the result is the PSNR. Adding other metrics would be advisable (e.g. Frechet Inception Distance).
- Having the number of parameters for baselines and competitors would be useful to compare them with the proposed method.
- For the reported experimental results, it is not clear if any of the proposed designs is better than the others.

### Comments
For future work, I would recommend the authors to focus on:
- Clarify how the denoising models can benefit from the usage of FFCs (e.g. training time, convergence stability, performance, overall results, better shape reconstruction, etc.).
- Investigate other deep architectures as base model and review if FFCs are beneficial in any of these models or only in specific ones (e.g. shallow/deep models, with/without skip connections, etc.).

**Suggested Changes:**

I think the paper does not require any significant changes.

I would suggest the authors to highlight the ares that are worth comparing in Fig. 1-3 and add a short discussion on the qualitative differences that they noticed between different methods (e.g. in Fig. 3, it seems that N2S is better at recovering colors while "Our best" is better at recovering shape. Is this true in other samples too?).
I also recommend to add the number of parameters for baselines and competitors for easier comparison with the proposed method.

---

### Author Response · Authors · 2023-05-31
**I wish to opt in for the archival**

I wish to opt in for the archival

---

### Meta-Review · Area_Chair_Cp8S · 2023-04-07

**Recommendation:** Invite to present
**Confidence:** 5

**Metareview:**

The motivation for using Fast Fourier Convolutions for image-denoising tasks is clear and backed by the experiments.

**Summary:**

The authors propose the use of Fast Fourier Convolutions to obtain a global view of the input image for image-denoising tasks. The experiments though show marginal improvement, showing the potential of using FFC for image-denoising tasks.

**Reason For Not Giving A Higher Recommendation:**

This paper is interesting and clearly written, the only concern I have is the experimental gains which ar marginal and could be the effect of random seeds as one of the reviewers noted. So I encourage the authors to run the statistically significant test.

**Reason For Not Giving A Lower Recommendation:**

na

---

### Decision · Program_Chairs · 2023-04-09

Invite to present